# Who Got Infected with COVID-19? A Study of College Students in Wuhan (China)

**DOI:** 10.3390/ijerph18052420

**Published:** 2021-03-02

**Authors:** Changyu Fan, Min Li, Xueyan Li, Miao Zhu, Ping Fu

**Affiliations:** School of Sociology, Central China Normal University, 152 Luoyu Avenue, Wuhan 430079, China; fanchangyu@mail.ccnu.edu.cn (C.F.); lizimin703@mails.ccnu.edu.cn (M.L.); lixueyan@mail.ccnu.edu.cn (X.L.); zhuzhu@mails.ccnu.edu.cn (M.Z.)

**Keywords:** COVID-19, college students, time of disease onset, infection status, group differences

## Abstract

College students represent a large group of people who frequently travel across regions, which increased their risk of infection and exacerbated the risk of COVID-19 spread throughout China. This study uses survey data from the end of April 2020 to analyze the status of COVID-19-infected cases, the group differences, and influencing factors in college students in Wuhan. The sample size was made up 4355 participants, including 70 COVID-19-infected students. We found that during the COVID-19 outbreak in early 2020, college students in Wuhan were primarily infected during off-campus events after winter break or infected in their hometowns after leaving Wuhan; the percentage of college students with severe cases was relatively low, and most had mild cases; however, a large proportion of asymptomatic cases may exist; there were significant group differences in gender, age and place of residence; and the risk of infection was closely related to the campus environment, in which the population density and number of faculty and students on campus had a significant impact. The results indicated that the infection of students did not occur at random, thus strengthening student health education and campus management can help curb the spread of COVID-19 among students.

## 1. Introduction

On 11 March 2020, the World Health Organization (WHO) declared the coronavirus disease 2019 (COVID-19) a pandemic [1].With large-scale populations traveling within China and around the world during the Spring Festival period, including a large number of college students and international students, COVID-19 spread across the globe [2,3,4]. Wuhan is one of the cities with the largest populations of college students in the world, with as many as 1.2 million. Such a large number of students returning home from college can not only increase their risk of infection but also exacerbate the risk of COVID-19 spreading throughout China and beyond. College students often have highly concentrated activities and frequent interactions on campus and travel across China in large numbers; therefore, after the outbreak, all colleges and universities across the country (all levels and all types) closed and instead, students attended online classes at home. As of the end of June 2020, no college has fully resumed classes in China. Globally, almost all countries that have originated COVID-19 cases have closed schools indefinitely during the pandemic. United Nations (UN) reports show that more than 1 billion students are out of school worldwide, and due to the global situation and the difficulty of change, decisions to return to school must consider public health, the benefits and risks of education and other factors [5].

The pandemic has exerted a greater impact than on college students than on elementary and middle school students. On the one hand, college students come from different regions in China or other countries, they are a group with huge cross-regional mobility. On campus, college students have more freedom in their studies and lifestyle. They don’t have a fixed place to study, and the optional nature of courses make many classrooms a temporary combination of different students. Furthermore, students have more time to go out after class and participate in activities in the city or other public places. These factors not only make them more susceptible to infection, but also make them a vector for virus transmission. On the other hand, home isolation during the epidemic has caused greater academic difficulties for college students. For students in some majors, special facilities, equipment, tests and on-site guidance are needed to complete experiments, training and data collection. Therefore, it is difficult to complete course learning only through online teaching, and the training of professional skills and academic are also affected. In this sense, the temporary college closure due to the pandemic may greatly reduce the quality of college students’ learning [6,7,8].

However, this is only one aspect of the matter. More importantly, there may be group differences in online-learning conditions among students, including network quality, learning devices, learning space and learning interruptions [9,10,11]. In China, there are significant urban-rural differences [12,13]. For example, the overall internet infrastructure in rural areas is not as good as in cities, and the networks used by rural households is slow and unstable. Students from rural areas may lack independent, comfortable learning space, may experience learning disruptions caused by family life, and may even need to engage in agricultural or other work activities to help family members. In other words, the long-term college closure may further expand the urban-rural inequality in education [14]. In addition, long-term studying at home and study pressure from various limiting factors could also bring anxiety, depression, and other psychological problems [15,16,17]. Therefore, discussing the characteristics and influencing factors of infected college students, we can carry out targeted epidemic prevention and health education for college students and strengthen campus health management, which is helpful to reduce the negative impact of epidemic situation on health education of college students.

As COVID-19 spreads around the world, research on student infection status is gradually increasing. Several monitoring studies based on universities in the United Kingdom and the United States show that the number of COVID-19 cases has increased rapidly on university campuses in the short term after the start of the fall semester in 2020 [18]. In terms of reasons, student gatherings and congregate living environments increasing the likelihood of rapid transmission [19,20]. In retrospect, during the severe acute respiratory syndrome (SARS) outbreak in Beijing in 2003, 175 students were infected, of which 90 were college students, accounting for more than half of the cases. During the SARS outbreak, there were three student deaths, all of them college students [21]. In addition to understanding the risk of infection among college students in Wuhan, this study aimed to determine the infection status of the resident population (confirmed cases) in Wuhan. Existing studies have shown that SARS-CoV-2 is highly infectious and has low mortality and COVID-19 can cause clusters of severe respiratory diseases [22]. According to report, at the end of 2019, Wuhan had more than 14 million residents, and 5 million people left Wuhan before the lockdown on 23 January 2020 [23]. Due to the large population movement in Wuhan, the group characteristics and identities of residents are ambiguous, which makes it extremely difficult to determine the infection rate (confirmed cases) of residents in Wuhan. College students represent a group with clear characteristics and boundaries. Through a scientific sampling survey, the general situation of SARS-CoV-2 transmission under the background of population movement can be understood.

In view of this, the evaluation and exploration of the COVID-19 infection status, group characteristics and influencing factors in college students are of great importance for students returning to campus, pandemic prevention and control, and reducing the negative impact of the epidemic.

## 2. Methods

### 2.1. Participants and Procedure

This study is one of the results of a large-scale survey: COVID-19 Impact Survey of Faculty and Students of Wuhan Universities and Colleges (CFSW). The college students from 83 higher education institutions (hereinafter referred to as colleges) in Wuhan were the respondents, and the survey was conducted from 26 to 29 April 2020. The survey was pre-approved by the Institutional Review Committee of the universities involved.

Using a multistage random sampling method, this survey aimed to recruit 4500 students. Specifically, first, 13 representative colleges were randomly selected from 83 colleges, and the number of students selected from each college was proportional to the number of enrollments in each college; second, 30% of the departments were randomly selected in each selected college, and the number of students selected from each department was proportional to the number of students in the department; finally, the sample pool in the department was established using the students’ identity (ID) number, and the corresponding students (n = assigned sample size) were randomly selected for the survey. We implemented college sampling, allocated the college-level sample sizes, designed the guidelines for the stratified random sampling procedure, and guided and assisted the student management department of each selected college to implement random sampling.

The survey used electronic questionnaires to collect data. Before the start of the survey, an informed consent statement was presented to selected students, and for students who agreed, the questionnaire was further administered in Chinese. It should be noted that in order to ensure a sufficient number of confirmed COVID-19 students, the management departments of the 13 selected colleges must report all the confirmed cases they know and include the cases in the investigation, namely, we over-sample the confirmed students. The survey was conducted with a total of 4355 participants, of which 70 students were confirmed with COVID-19.

### 2.2. Measurements

The questionnaire measured infection, diagnosis, and treatment in students through a series of questions. In the questionnaire, all the students were asked about the specific time when they left Wuhan for their hometowns during winter break in January 2020. Additionally, the students were also asked the following question: “During the pandemic, did you have following symptoms? (1) itchy throat and dry cough, (2) weakness and joint pain, (3) fever, (4) confirmed influenza, (5) pneumonia but not confirmed COVID-19 infection, and (6) confirmed COVID-19 infection”. If the respondent answered “(6) confirmed COVID-19 infection”, then the time of disease onset, disease severity, treatment time, physical condition, etc. were addressed in subsequent questions. Specifically, the infected students were asked for time of disease onset, which was combined with the time of departure from Wuhan to describe the prevalence of COVID-19 among college students in Wuhan. Disease severity during the pandemic was divided into mild, severe, critical and asymptomatic; the treatment time was divided into “<10 days”, “10–20 days”, “>20 days”, and “not cured”; and the self-evaluation of physical condition at the end of the questionnaire included the responses “very good”, “good”, “average”, “poor”, and “very poor”.

The variables for sociodemographic characteristics included gender, birth cohort, type of household registration, education level, major, years in college, and nature of college attended. Among them, the age is divided into below 20 years old, 20–25 years old, 25 years old and above; household registration type included city, township, and rural area; education level included junior college, undergraduate, master and doctor; major included liberal arts and science and engineering; years in college included 1 year, 2 years, 3 years, and 4 years and above; and nature of college attended included college directly affiliated with the Ministry of Education, provincial or municipal college, private college, and provincial or municipal junior college. The student was also asked if he/she is a member of the Communist Party of China (CPC) and if he/she would graduate this semester.

To analyze the factors that influence COVID-19 infection in students, some variables related to virus transmission were also introduced, including place of residence during the pandemic (January 23 to April 8, 2020) and severity of the COVID-19 pandemic, which included Wuhan, other cities or counties in Hubei Province, and other provinces. Baidu maps (https://map.baidu.com/@12735205,3551462,13z, accessed on 20 May 2020) was used to measure the walking distance between each surveyed college (the nearest campus) and the Huanan South China Seafood Market (hereinafter referred to as the distance to Huanan South China Seafood Market (km). The 2019 data for the faculty and students at each college surveyed were obtained from the education administration department of Hubei Province, building area data were collected from the official websites of the surveyed colleges, and 2 variables based on these data were established: size (1000 people) and population density (100 people/10,000 m^2^) of the faculty and students at each college. Table 1 shows a statistical description of the variables involved in the regression analysis.

### 2.3. Statistical Analyses

Descriptive statistical analysis was performed on the infection rate (confirmed cases), time of disease onset, disease condition, and recovery status of infected students, and the characteristics of uninfected and infected students were compared. To further clarify which groups of students are more susceptible to COVID-19 infection (confirmed cases), this study adopted a logit regression model for analysis. The number of infected students was very small, i.e., a rare event, which could lead to estimation bias (rare event bias) when using maximum likelihood estimation (such as Logit or Probit). Therefore, complementary log-log regression was used to correct the rare event bias.

Among the 4355 students in this study, 3825 (87.39%) left Wuhan, with most returning to their hometowns during winter break (students who left Wuhan to go to their hometowns). All students and students who left Wuhan to go to their hometowns were the 2 sample data sets for the regression analysis, and date of departure from Wuhan (the last date is 24 January 2020) was added in the regression to analyze the relationship between the time students left Wuhan and confirmed COVID-19 infection.

Because the infected students were not selected through sampling, it was not appropriate to directly combine these students with uninfected students. Some colleges increased their sample size, and the descriptive statistics showed that the sample size of the students in lower grades was higher. Therefore, based on the overall number of college students in Wuhan, sampling weights were calculated using infection status (confirmed cases), college type and college years and were used in the regression model to correct sample bias.

## 3. Results

### 3.1. Infection Status (Confirmed Cases)

Among the 4355 students surveyed, there were 70 infected students (no deaths occurred). A total of 290,967 students enrolled in the 13 colleges; therefore, the calculated infection rate (confirmed cases) for college students in Wuhan was 1.61‰, and the 95% confidence interval (CI) was [1.26‰, 2.01‰].

In addition to the infected students, 287 (6.7%) of the 4285 uninfected students showed relevant symptoms during the pandemic (January 23–April 8, 2020). Among them, 91 students (2.12%) had an itchy throat and dry cough, 23 students (0.54%) had body weakness and joint pain, 61 students (1.42%) had fever, 178 students (4.15%) had influenza, and three students (0.07%) had pneumonia but not confirmed COVID-19. For the infected students, the survey showed that family members of 38 students were confirmed with COVID-19; for the uninfected students, family members of 19 students were confirmed.

### 3.2. Time of Disease Onset

Figure 1 shows the epidemiological curves for the date students left Wuhan to go to their hometowns and the date of disease onset. It can be seen from the figure that during the COVID-19 pandemic, the peak number of students left Wuhan on 10 January 2020, which was the last day of the final week for most colleges in Wuhan. By the end of January 10, 2020, 2372 students (62.32%) had left Wuhan, and by 23 January 2020, 3651 students (95.93%) had left Wuhan. In other words, almost all students who were returning to their hometowns had left Wuhan.

Seventy infected students were included in the survey. Figure 1 shows that the first infected college student experienced disease onset on 18 January 2020, just 7 days after the first peak. 24 January 2020, was the date with the peak number of infected students, with six students reporting disease onset on that day, just 14 days after the first peak. Overall, for the date students left Wuhan to go to their hometowns and the date of disease onset, a two-peak distribution, 14 days apart, was formed. Existing studies have indicated that the incubation period of COVID-19 is generally 3–7 days, with a maximum of approximately 14 days [24]. It can be inferred that most of the infected students were infected when they left campus for vacation or went out for activities. Notably, 14 days after the lockdown of Wuhan, new cases were still emerged between 14 February and 21 February 2020, and there was even one case on 23 April.

Figure 2 shows the gap between the time of departure from Wuhan to the time of disease onset in students. For the 38 infected students who left Wuhan, the date of disease onset was before the date of departure from Wuhan for four infected students, the date of disease onset was within 1 week after leaving Wuhan for 17 infected students (the highest number, 44.74%), and the date of disease onset was within 1–2 weeks after leaving Wuhan for 9 infected students (23.68%). In addition, for eight infected students (21.05%), the date of disease onset was over 2 weeks after leaving Wuhan, and among them, seven students lived in other cities and counties in Hubei Province, where the pandemic was also serious.

### 3.3. Symptoms of Disease

Table 2 shows that among the 70 infected students, 59 (84.29%) had mild cases and 5 (7.14%) had severe and critical cases. In addition, six students (8.57%) had asymptomatic cases. In terms of the time of disease onset, before the lockdown of Wuhan on 23 January 2020, the rate of severe/critical cases in infected students was very high, reaching 12.5%; in the first 14 days of late January to early February, the rate decreased to 7.5%. There was no severe/critical case after February 7.

### 3.4. Recovery Status

Table 3 and Table 4 show the recovery status of the infected students. In terms of treatment time, 39 students (55.71%) required longer than 20 days, including all case types, 23 students (32.86%) required 10–20 days, and a small number patients with mild and asymptomatic cases were cured within 10 days; only one student, who had a critical case, was not completely cured (Table 3). Self-evaluation of physical condition showed that, overall, the recovery of infected students was relatively good, even for those with severe/critical cases. Among them, 34 students (48.57%) felt that they were in very good physical condition, 30 students (42.86%) felt that they were in good condition, and 5 students (7.14%) felt that they were in average condition; the critically ill student, who was not fully cured, felt that he/she was in very poor physical condition (Table 4). This student originated disease onset on 1 February 2020, and had been receiving treatment for nearly 3 months by the end of the survey (Figure 1).

### 3.5. Group Characteristics

In Table 5, the chi-square test results show that except for member of the CPC, major and years in college, there were significant differences between the infected and uninfected students; specifically, in terms of demographic characteristics, male students accounted for a significantly larger proportion (67.14% vs. 50.46%, *p* < 0.05); older students were the majority; the percentage of students born before 1995 was significantly higher (20% vs. 4.18%, *p* < 0.001); and the proportion of students whose household registration type is city was significantly higher (62.86% vs. 26.09%, *p* < 0.001).

In terms of educational characteristics, the infection rates in graduate students (32.86% vs. 9.66%) and Ph.D. students (8.57% vs. 2.59%) (*p* < 0.001) were significantly higher, and they were concentrated in the colleges directly affiliated with the Ministry of Education (57.14% vs. 24.32%); additionally, the percentage of students in the graduating class (25.71% vs. 14.28%, *p* < 0.05) also exceeded the corresponding proportion by approximately 10%. Colleges directly affiliated with the Ministry of Education have higher education levels and school-administration levels than do provincial colleges in Hubei, and all offer education programs ranging from the undergraduate level to the master’s and doctoral levels; for provincial colleges in Hubei, only 15% offer master’s programs, and 8% offer Ph.D. programs. As a result, students enrolled in colleges directly affiliated with the Ministry of Education, especially master’s and doctoral students, are more likely to stay in Wuhan during winter break to engage in scientific research, and they are more likely to return to their hometowns at later time, thus facing a higher risk of infection.

### 3.6. Influencing Factors

To further analyze the factors that influence the infection rate (confirmed cases) in college students in Wuhan, we used the logit model and complementary log-log model for regression analysis. However, as presented in Table 6, comparing AIC and BIC, it is found that the difference in the goodness of fit between the two models is very small.

In model 1, in terms of individual characteristics, there were significant differences in the three demographic variables, i.e., gender, age, and type of household registration; however, there was no significant differences in the educational characteristics of students. Specifically, after controlling other variables, the infection rate (confirmed cases) for male students was only 0.52 times that for female students (95% CI [0.29, 0.94], *p* < 0.05). The effect of age was also significant; that is, on average, the infection rate increased by 21% for every 1-year increase in age (95% CI [6%, 39%], *p* < 0.05). There were also significant urban-rural differences in the infection rate, namely, the infection rate for students with an urban registration (city) was higher than that for students with a rural registration by 181% (95% CI [51%, 421%], *p* < 0.05). Among the students, there was a significant difference between CPC members and non-CPC members, and the infection rate for CPC members was 0.37 times higher than that for non-CPC members (95% CI [0.19, 0.71], *p* < 0.05).

The regression results showed that college characteristics, such as types of college, number of faculty and students, and population density of faculty and students on campus, had a significant impact on the infection rate (confirmed cases), but the effect of the distance to Huanan South China Seafood Market was not significant. Specifically, students in colleges directly affiliated with the Ministry of Education had a significantly higher infection rate (*p* < 0.05) than did students in the other three types of colleges, whose infection rate were 0.01 times that of students in colleges directly affiliated with the Ministry of Education. Because SARS-CoV-2 transmission relies primarily on interaction among people, the population density and the number of faculty and students on campus are important factors that influence virus transmission. The analysis results showed that the population density of faculty and students on campus had a significant positive impact on the infection rate for students and that an increase of 100 people per 10,000 m^2^ could increase the risk of infection by 113% (95% CI [27%, 257%], *p* < 0.05). The squared term of the number of students and faculty in college was added to the model, and the analysis results showed that the effects of both the number of students and faculty and the squared term were significant (*p* < 0.05), which means that there is a “U”-type relationship between the number of students and faculty and the infection rate for students, indicating that the risk of infection is significantly higher in colleges with small and large sizes of faculty and students.

Finally, the impact of activities during the pandemic on the infection rate (confirmed cases) was analyzed. Model 1 showed that the risk of infection among students living in Wuhan during the pandemic was significantly higher than that of students living outside Wuhan. The risk of infection among students living in other cities and counties in Hubei Province was only 0.25 times that of those living in Wuhan (95% CI [0.23, 0.50], *p* < 0.001), and risk of infection for students living in other provinces was only 0.18 times that for those living in Wuhan (95% CI [0.1, 0.33], *p* < 0.001).

Model 3 and model 4 specifically analyzed the impact of the time of departure from Wuhan on the infection rate. After controlling other variables, model 4 showed that the later the date of departure from Wuhan and the closer the date to 23 January 2020 (the lockdown of Wuhan), the significantly higher the infection rate was; that is, leaving Wuhan 1 day later could cause the infection rate to increase by 16% (95% CI [9%, 23%], *p* < 0.001). This result is consistent with the spread trends for SARS-CoV-2. In addition, among students who left Wuhan to go to their hometowns, the gender difference of infection disappeared, but the difference in education level become very significant, compared with junior college students, the infection rate of undergraduates, masters and doctoral students increased by 4–11 times.

## 4. Discussion

Statistics showed that there were 30.32 million college students in China in 2019 [25], and they traveled all over the country (including remote rural areas, large and medium-sized cities). There were also groups of international students who migrated across the country, which form a mobile social network around the world. This undoubtedly creates favorable conditions for the spread of the virus. In other words, restricting the unnecessary movement of undergraduates is vital for epidemic prevention. However, under the global situation of the pandemic, it is very likely that SARS-CoV-2 will coexist with human beings for a long time, and the epidemic may occur repeatedly in the fall, winter or spring every year [26]. Therefore, it is unrealistic to adopt the long-term implementation of school closure or suspension measures, and how to construct an orderly flow of college students is the key to the education department to conduct epidemic prevention in campus.

As of 17 May 2020, 50,339 people were confirmed in Wuhan, and the infection rate (confirmed cases) was 5.6‰ based on a population of 9 million [27]. In this survey, the infection rate (confirmed cases) for college students in Wuhan was 1.61‰, so it may greatly overstate the infection rate of students. Presuming from the actual situation, the infection rate of students should be lower than that of the residents. First, college students are usually between the ages of 18 and 30. They are young and healthy and have strong immunity; therefore, the risks of infection or disease onset are lower than other populations. A study found that among 72,314 COVID-19 cases from mainland China, only 13.4% of the confirmed cases were under 30 years old [28]. Second, the activities of the vast majority of college students in the city are limited and in relatively closed areas, mainly on campus. When the pandemic broke in Wuhan at the end of December 2019, college students in Wuhan were reviewing their courses and preparing for final exams in dormitories or classrooms; therefore, the scope of student activities is smaller than usual. Third, as the Chinese Spring Festival approached, the largest population movement in China occurred, which may have led to the widespread diffusion of the pandemic. However, most of the students had already returned to their hometowns before the Spring Festival, and the risk of infection was greatly reduced.

According to studies, the incubation period of infection is generally 3–7 days, with an approximate upper limit of 14 days [24]. Therefore, it can be inferred that the students were infected during winter break. In other words, during winter vacation, students had more off-campus activities, leading to an increase in the infection rate. In addition, the gap between the date of outbreak and the date of departure from Wuhan also showed that 21.05% of the infected students originated disease onset >2 weeks after leaving Wuhan. Given this large proportion, it is reasonable to assume that the infection occurred when students returned home, because the typical incubation period is much shorter than 2 weeks.

Some studies indicated that, as of 11 February 2020, four-fifths of COVID-19-infected patients in China had mild to moderate symptoms and the percentage of severe/critical cases was 18.5% [25]. This survey showed that 84.29% of the infected students had mild symptoms and the severe/critical cases were 7.14%, less than half the national average. The results indicated that the high rate of severe/critical cases in the early stage of the epidemic may be due to shortage of medical resources. Notably, 8.57% of the infected students had asymptomatic cases. The infection rate (confirmed cases) for college students in Wuhan was much lower than that of Wuhan residents and populations without immunity and generally susceptible to SARS-CoV-2 [29]; therefore, there may be more asymptomatic cases in students. An outbreak of COVID-19 among University of Texas (Austin, TX USA) students returning from spring break occurred in March 2020, and of the 64 infected college students and their close contacts, 14 were asymptomatic, accounting for 22% [30], which supports the figures in our study. Therefore, from the perspective of pandemic prevention, it is necessary to conduct a pathogenetic examination of students before returning to college. In view of the unknown sources of infection among students, antibody screening is required, and centralized isolation and medical observation of patients with asymptomatic cases are also necessary [31].

The comparison of group characteristics showed that infection in students did not occur randomly and there were obvious group differences, especially in gender, age, type of household registration, date of departure from Wuhan, and place of residence during the pandemic. In other words, the individual factors that affect COVID-19 infection in students are mainly physiological and life-related characteristics. The result is consistent with the analysis of infected cases among residents; that is, most infected patients are middle-aged and elderly. Although there is no obvious gender difference in infected patients, the mortality rate for males is as high as 50%, and the infection rate (confirmed cases) for those living in Hubei and Wuhan is even higher [28].

Finally, some surveys have shown that clustered outbreaks can easily occur in enclosed locations with large numbers of people [32,33]. On campus, classrooms, libraries, and canteens are places with high population densities and frequent interpersonal interactions and are the focus of pandemic prevention and control. In these places, it is not only necessary to wear masks and maintain social distancing, but also to instruct students on the methods of social activities during the pandemic to minimize the risk of infection. The analysis of the number of faculty and students showed that there was a “U”-shaped relationship between the number of faculty and students, and the infection rate (confirmed cases) of students, which means that the risk of infection was significantly higher in colleges with a large numbers of faculty and students. It may be related to the social interaction distance. In colleges with large numbers of faculty and students, the frequency of interpersonal interaction may be high. It is worth noting that the distance to Huanan South China Seafood Market did not have a significant impact on the infection rate (confirmed cases) for students. To a certain extent, it indicates that the spread of viruses among students is not related to the original outbreak, the Huanan South China Seafood Market, but may be related to the locations of students’ off-campus activities. It further indicates that it is necessary to restrict students from leaving campus and adopt closed management.

One of the main advantages of our study is that the colleges in Wuhan cooperated well with large-scale, rigorous sampling survey of students to ensure unbiased statistical results. However, some limitations of the present study should be considered while interpreting the findings. First, although the survey design obviously needs to investigate all infected students, students may have been missed due to various reasons during the process. For example, students may hide their illness and never report it in the survey. In addition, since the students did not return to college, the electronic questionnaires were completed by themselves. Therefore, the authenticity of the responses could not be fully guaranteed because there was no monitoring by investigators. Third, in an epidemiological survey, there may be recall bias, especially at the onset of disease. As an important analytical variable, the results of onset time may be biased. Due to the aforementioned limitations, some results of this study should be interpreted with caution (such as the infection rate).

## 5. Conclusions

The main findings in this study are as follows: (1) college students were primarily infected during off-campus events after winter break or infected in their hometowns after leaving Wuhan; (2) among the college students, the percentage of those with severe cases was relatively low, and most had mild cases; however, a large proportion of asymptomatic cases may exist; (3) the individual factors that affect infection are mainly demographic factors, such as gender, age and place of residence, while educational characteristics are not closely related to infected students; and (4) the risk of infection was closely related to the campus environment, in which the population density and number of faculty and students on campus had a significant impact.

Therefore, from the perspective of pandemic prevention and control on campuses, our study suggests that first, because a high proportion of students may have asymptomatic infections, antibody tests should be required before returning to college, and students with asymptomatic cases should be isolated and treated; second, during the pandemic, in order to minimize off-campus activities and to conduct certain tracking and management of students who leave campus, colleges should implement closed management after they are reopened; finally, social distancing should be maintained on campus, especially for colleges with a large population density of faculty and students (which is calculated based on building area for each college) and colleges with a large numbers of faculty and students.

## Figures and Tables

**Figure 1 ijerph-18-02420-f001:**
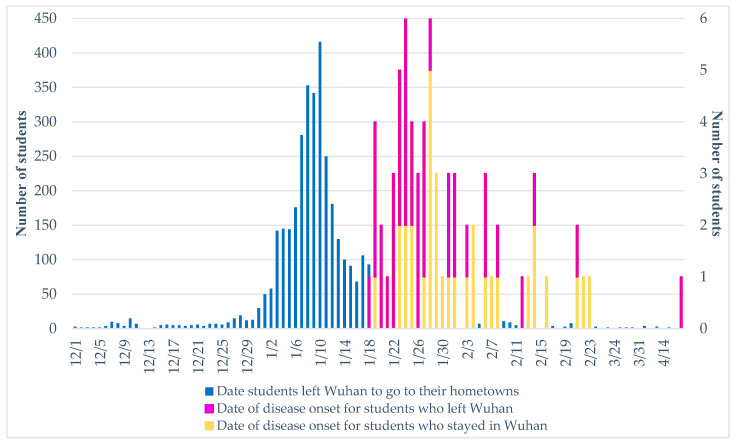
Epidemiological curves for the date students left Wuhan to go to their hometowns and the date of disease onset.

**Figure 2 ijerph-18-02420-f002:**
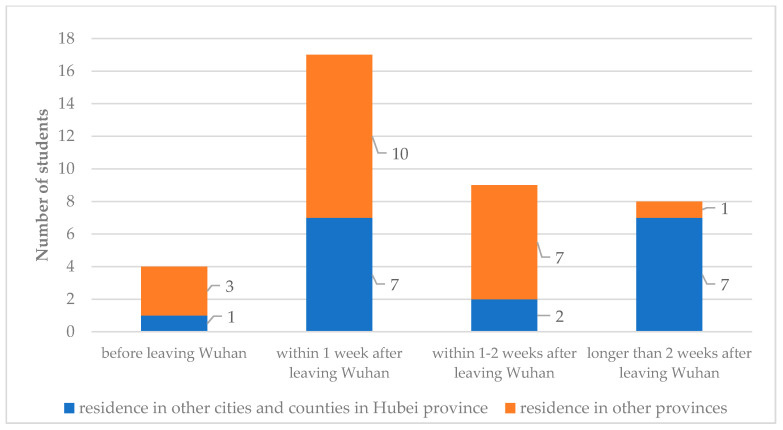
Distribution of the gap between time of departure from Wuhan to time of disease onset in students (*n* = 38).

**Table 1 ijerph-18-02420-t001:** Variable statistical description (*n* = 4355).

Variable	Measurement	*n* (%)/Mean (SD)
COVID-19 infection	No	4285 (98.39)
	Yes	70 (1.61)
Gender	Male	2209 (50.72)
	Female	2146 (49.28)
Age	-	20.87 (2.22)
Household registration type	Rural area	2405 (55.22)
	Township	788 (18.09)
	City	1162 (26.68)
Member of CPC	No	3590 (82.43)
	Yes	765 (17.57)
Education level	Junior college	1683 (38.65)
	Undergraduate	2118 (48.63)
	Master’s	437 (10.03)
	Doctor	117 (2.69)
Years in college	1 year	2152 (49.41)
	2 years	1199 (27.53)
	3 years	664 (15.25)
	4 years and above	340 (7.81)
College type	college directly affiliated with the Ministry of Education	1082 (24.85)
	Provincial college	1340 (30.77)
	Private college	450 (10.33)
	Provincial junior college	1483 (34.05)
Distance to Huanan South China Seafood Market (km)	-	18.49 (8.71)
Population density of faculty and students on campus (100 persons/10,000 m^2^)	-	3.34 (1.12)
Number of faculty and students (1000 persons)	-	28.81 (19.36)
The squared term of the number of students and faculty	-	1205.06 (1517.76)
Residence during the pandemic	Wuhan	657 (15.09)
	Other cities and counties in Hubei Province	2087 (47.92)
	Other provinces	1611 (36.99)
Date of departure from Wuhan to hometown (Date)	-	9.94 (6.01)

**Table 2 ijerph-18-02420-t002:** Symptoms of disease in infected students.

Time of Disease Onset	Mild (%)	Severe (%)	Critical (%)	Asymptomatic (%)
Before 23 January 2020 (*n* = 16)	14 (87.5)	2 (12.5)	0 (0.00)	0 (0.00)
24 January 2020–6 February 2020 (*n* = 40)	33 (82.50)	2 (5.00)	1 (2.50)	4 (10.00)
After 7 February 2020 (*n* = 14)	12 (85.71)	0 (0.00)	0 (0.00)	2 (14.29)
Total (*n* = 70)	59 (84.29)	4 (5.71)	1 (1.43)	6 (8.57)

Note: During the COVID-19 pandemic from January to April 2020, China’s National Health Commission issued “Chinese Clinical Guidance for COVID-19 Pneumonia Diagnosis and Treatment (Trial 7th Edition)” (Available online: http://www.nhc.gov.cn/yzygj/s7653p/202003/46c9294a7dfe4cef80dc7f5912eb1989.shtml, accessed on 4 March 2020). According to which the hospital classified the infected people into “Light, Moderate, Severe and Critically type”, and confirmed that the infected with “asymptomatic” could also be the source of infection. In the questionnaire, we requested the infected students to report their diagnosis and treatment results. We merge the “Light” and “Moderate” together as “Mild”.

**Table 3 ijerph-18-02420-t003:** Treatment time for infected students.

Severity of Disease	<10 Days (%)	10–20 Days (%)	>20 Days (%)	Not Cured (%)
Mild (*n* = 59)	5 (8.47)	21 (35.59)	33 (55.93)	0 (0.00)
Severe (*n* = 4)	0 (0.00)	0 (0.00)	4 (100.00)	0 (0.00)
Critical (*n* = 1)	0 (0.00)	0 (0.00)	0 (0.00)	1 (100.00)
Asymptomatic (*n* = 6)	2 (33.33)	2 (33.33)	2 (33.33)	0 (0.00)
Total (*n* = 70)	7 (10.00)	23 (32.86)	39 (55.71)	1 (1.43)

**Table 4 ijerph-18-02420-t004:** Physical conditions of infected students by self-evaluation.

Severity of Disease	Very Good (%)	Good (%)	General (%)	Poor (%)	Very Poor (%)
Mild (*n* = 59)	28 (47.46)	27 (45.76)	4 (6.78)	0 (0.00)	0 (0.00)
Severe (*n* = 4)	1 (25.00)	2 (50.00)	1 (25.00)	0 (0.00)	0 (0.00)
Critical (*n* = 1)	0 (0.00)	0 (0.00)	0 (0.00)	0 (0.00)	1 (100.00)
Asymptomatic (*n* = 6)	5 (83.33)	1 (16.67)	0 (0.00)	0 (0.00)	0 (0.00)
Total(*n* = 70)	34 (48.57)	30 (42.86)	5 (7.14)	0 (0.00)	1 (1.43)

**Table 5 ijerph-18-02420-t005:** Characteristics of infected and uninfected students in Wuhan.

Variable	Measurement	Uninfected Students (*N* = 4285) *n* (%)	Infected Students (*N* = 70) *n* (%)	Chi-Square Test *p* Value
Gender	Male	2162 (50.46)	47 (67.14)	0.006
	Female	2123 (49.54)	23 (32.86)	
Birth cohort	Before 1995	179 (4.18)	14 (20.00)	0.000
	1995–1999	1694 (39.53)	34 (48.57)	
	After 2000	2412 (56.29)	22 (31.43)	
Type of household registration	Rural area	2386 (55.68)	19 (27.14)	0.000
	Township	781 (18.23)	7 (10.00)	
	City	1118 (26.09)	44 (62.86)	
Member of the CPC	No	3534 (82.47)	56 (80.00)	0.590
	Yes	751 (17.53)	14 (20.00)	
Education level	Junior college	1660 (38.74)	23 (32.86)	0.000
	Undergraduate	2100 (49.01)	18 (25.71)	
	Master	414 (9.66)	23 (32.86)	
	Doctor	111 (2.59)	6 (8.57)	
Type of major	Liberal arts	1933 (45.11)	28 (40.00)	0.394
	Science and engineering	2352 (54.89)	42 (60.00)	
Years in college	1 year	2126 (49.61)	26 (37.14)	0.231
	2 years	1175 (27.42)	24 (34.29)	
	3 years	651 (15.19)	13 (18.57)	
	4 years and above	333 (7.77)	7 (10.00)	
Graduating class	No	3673 (85.72)	52 (74.29)	0.007
	Yes	612 (14.28)	18 (25.71)	
College type	College directly affiliated with the Ministry of Education	1042 (24.32)	40 (57.14)	0.000
	Provincial college	1334 (31.13)	6 (8.57)	
	Private college	448 (10.46)	2 (2.86)	
	Provincial junior college	1461 (34.10)	22 (31.43)	

**Table 6 ijerph-18-02420-t006:** The robust regression analysis of logit and cloglog of confirmed infection among college students in Wuhan.

Dependent Variable: Infected or Uninfected (Infected = 1)	All Students		Students who Left Wuhan	
	Model 1: Logit	Model 2: Cloglog	Model 3: Logit	Model 4: Cloglog
Gender (Male = 0)	0.519 ** [0.285,0.944]	0.524 ** [0.298,0.922]	0.766 [0.347,1.691]	0.756 [0.352,1.624]
Age	1.213 ** [1.058,1.390]	1.190 ** [1.059,1.337]	1.232 ** [1.045,1.452]	1.221 ** [1.052,1.417]
Household registration type (Rural area = 0)				
Township = 1	1.087 [0.444,2.660]	1.088 [0.453,2.615]	1.424 [0.522,3.885]	1.403 [0.525,3.748]
City = 2	2.808 ** [1.513,5.212]	2.737 ** [1.498,5.003]	2.739 ** [1.321,5.677]	2.637 ** [1.296,5.364]
Member of CPC (No = 0)	0.367 ** [0.189,0.714]	0.388 ** [0.203,0.742]	0.375 ** [0.178,0.793]	0.397 ** [0.193,0.819]
Education level (Junior college = 0)				
Undergraduate = 1	1.071 [0.256,4.487]	0.983 [0.234,4.130]	10.76 *** [2.712,42.705]	10.46 *** [2.655,41.205]
Master’s = 2	1.481 [0.311,7.048]	1.332 [0.285,6.215]	12.93 *** [2.911,57.380]	12.62 **^*^ [2.928,54.412]
Doctor = 3	0.843 [0.130,5.467]	0.808 [0.129,5.055]	4.912 ** [1.089,22.148]	4.858 ** [1.114,21.185]
Years in college (1 year = 0)				
2 years = 1	1.461 [0.773,2.763]	1.425 [0.764,2.656]	1.376 [0.570,3.321]	1.342 [0.562,3.203]
3 years = 2	0.988 [0.393,2.487]	0.986 [0.406,2.395]	0.968 [0.255,3.680]	1.018 [0.287,3.617]
4 years and above = 3	1.484 [0.441,4.991]	1.475 [0.465,4.681]	1.399 [0.311,6.292]	1.495 [0.359,6.219]
College type (college directly affiliated with the Ministry of Education =0)				
Provincial college = 1	0.00257 ** [0.000,0.171]	0.00320 ** [0.000,0.177]	0.00000482 ** [0.000,0.034]	0.00000519 ** [0.000,0.035]
Private college = 2	0.000273 ** [0.000,0.117]	0.000348 ** [0.000,0.116]	0.00000112 ** [0.000,0.025]	0.00000119 ** [0.000,0.023]
Provincial junior college = 3	0.00362 ** [0.000,0.627]	0.00406 ** [0.000,0.595]	0.0000322 ** [0.000,0.289]	0.0000339 ** [0.000,0.299]
Distance to Huanan South China Seafood Market (km)	1.002 [0.946,1.060]	1.001 [0.948,1.058]	1.005 [0.894,1.130]	1.005 [0.895,1.129]
Population density of faculty and students on campus (100 persons/10,000 m^2^)	2.130 ** [1.268,3.576]	2.115 ** [1.255,3.566]	4.988 ** [1.189,20.935]	4.977 ** [1.173,21.125]
Number of faculty and students (1000 persons)	1.000 *** [0.999,1.000]	1.000 *** [0.999,1.000]	0.999 *** [0.999,1.000]	0.999 *** [0.999,1.000]
The squared term of the number of students and faculty	1.005 *** [1.002,1.007]	1.005 *** [1.002,1.007]	1.009 *** [1.004,1.014]	1.009 *** [1.004,1.014]
Residence during the pandemic (Wuhan = 0)				
Other cities and counties in Hubei Province = 1	0.250 *** [0.126,0.496]	0.274 *** [0.138,0.545]		
Other provinces = 2	0.180 *** [0.098,0.331]	0.199 *** [0.111,0.355]		
Date of departure from Wuhan to hometown (Date)			1.163 *** [1.091,1.240]	1.161 *** [1.090,1.236]
*N*	4355	4355	3825	3825
pseudo *R*^2^	0.233	—	0.271	—
Log pseudolikelihood	−275.1589	−75.3885	−155.3576	−155.1204
AIC	596.3178	596.777	354.7152	354.2407
BIC	743.0366	743.4959	492.2001	491.7256

Note: Odds Ratio [95%Confidence Intervals], ** *p* < 0.05, *** *p* < 0.001.

## Data Availability

The data presented in this study are available on request from the corresponding author. The data are not publicly available due to privacy restrictions.

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
