# Peer review of "Who Got Infected with COVID-19? A Study of College Students in Wuhan (China)"

_ijerph, 2021, doi:10.3390/ijerph18052420_

Round 1
Reviewer 1 Report
1.
2.2. Measurements
“Disease severity during the pandemic was divided into mild, severe, critical and asymptomatic;”
Did the level of symptom confirm by hospitals? Or did they have any guideline?
2.
Table 1. Variable statistical description (n=4355).
“COVID-19 infection Yes/ No” seems label wrong.
3.
The author tried to use different models to solve the bias. It’s nice.
Discussion section is too long with some repeat concepts. Please reduce overlap part in this section.
Author Response
Review report form from Reviewer 1
- 2. Measurements
“Disease severity during the pandemic was divided into mild, severe, critical and asymptomatic;”
Did the level of symptom confirm by hospitals? Or did they have any guideline?
Response: As stated on Page 7, Line 426-432.
During the COVID-19 pandemic from January to April 2020, China’s National Health Commission issued “Chinese Clinical Guidance for COVID-19 Pneumonia Diagnosis and Treatment (Trial 7th Edition)”. (Available online: http://www.nhc.gov.cn/yzygj/s7653p/202003/46c9294a7dfe4cef80dc7f5912eb1989.shtml (accessed on 4 March 2020). According to which the hospital classified the infected people into "Light, Moderate, Severe and Critically type", and confirmed that the infected with "asymptomatic" could also be the source of infection. In the questionnaire, we requested the infected students to report their diagnosis and treatment results. We merge the "Light" and "Moderate" together as "Mild" and added a note under Table 2.
- Table 1. Variable statistical description (n=4355).
“COVID-19 infection Yes/ No” seems label wrong.
Response: Thank you very much for your reminder, and we have modified the typos as NO/YES in Table 1(Page 4, Line 334).
- The author tried to use different models to solve the bias. It’s nice.
Discussion section is too long with some repeat concepts. Please reduce overlap part in this section.
Response: Thank you very much for your kind suggestion, and we have made the discussion part more concise. As stated on Page 11-13(Please refer to the content highlighted in yellow in the delete tag).

Reviewer 2 Report
It is a very interesting paper but some points needs to be checked out
An English native speaker review is mandatory because in the present form there are different passages not clear.
Please provide to check all the Reference in the text. Square bracket or apex ? (e.g. reference 19 in the text)
The reference list must be correct. Sci-hub is not an official sources (e.g. N° 19 and N° 23)
In the Introduction Section, the Authors mentioned Sci-Hub database, please clarify because sci-hub is a pirate web site. Please clarify how you conducted analysis.
Could be interesting if the Authors could specified how many students attending the classroom in order to better understand the specific density.
Could be interesting if the Authors could specified which are the urban or rural area interested in the migration phenomenon of the students in order to compare the datas of contagion.
Please clarified also what are the symptoms that you use to classified mild, severe and critical cases.
Discussion section is too long. The Authors some times repeat the same concept. Please reduce this section.
Author Response
Review report form from Reviewer 2
It is a very interesting paper but some points needs to be checked out
- An English native speaker review is mandatory because in the present form there are different passages not clear.
Response: Thank you very much for your reminder. We try our best to modify, if still not meet the requirements, we will find professional organization to deal with it when time permits.
(In the article, we have made all corrections using revision mode)
- Please provide to check all the Reference in the text. Square bracket or apex ? (e.g. reference 19 in the text)
Response: Thank you very much for your suggestion. We carefully checked all the References and adopted Square bracket uniformly in the article.
- The reference list must be correct. Sci-hub is not an official sources (e.g. N° 19 and N° 23)
In the Introduction Section, the Authors mentioned Sci-Hub database, please clarify because sci-hub is a pirate web site. Please clarify how you conducted analysis.
Response: Thank you very much for your suggestion. We carefully checked all the References and deleted non-standard citations, for example in page 1, line 31-32, “Wuhan has one of the largest populations of college students in the world, with as many as 1.2 million.”[4] “The data was provided by Hubei Provincial Department of Education” was deleted.
As for reference [19] in the text, We re-searched in PubMed (students infected with COVID-19) and found that there were already papers evaluating for infected students. We deleted the search results using the search engine and added new documents.
The statement has been modified in the text: Page 2, Line 92-97.
As COVID-19 spreads around the world, researches on the status of infected students are gradually increasing. Several monitoring studies based on British and American universities show that COVID-19 cases are rapidly increasing on university campuses in the short term after the start of the fall semester of 2020[18]. In terms of reasons, student gatherings and congregate living settings increase the likelihood of rapid spread of the virus [19,20].
Here are three of the references:
[18] Salvatore, Phillip P et al. Recent Increase in COVID-19 Cases Reported Among Adults Aged 18-22 Years - United States, May 31-September 5, 2020. Morbidity and mortality weekly report. 2020, 69,39 1419-1424. doi:10.15585/mmwr.mm6939e4.
[19] Wilson E, Donovan CV, Campbell M, et al. Multiple COVID-19 Clusters on a University Campus - North Carolina, August 2020. Morb Mortal Wkly Rep. 2020;69(39):1416-1418. doi:10.15585/mmwr.mm6939e3.
[20] Office for National Statistics. How has coronavirus (COVID-19) spread among students in England? Available online: https://www.ons.gov.uk/peoplepopulationandcommunity/educationandchildcare/articles/howhascoronaviruscovid19spreadamongstudentsinengland/2020-12-21. (Accessed on 10 February 2021).
- Could be interesting if the Authors could specified how many students attending the classroom in order to better understand the specific density.
Response: Sorry that we did not investigate the student density in the classroom. In fact, on the one hand, the classroom of college students is not fixed, and it is difficult for students to accurately report the density information such as the area of classroom and number of students. On the other hand, we can only get the building area information of the school through the college website.
- Could be interesting if the Authors could specified which are the urban or rural area interested in the migration phenomenon of the students in order to compare the datas of contagion.
Response: The phenomenon of student migration mentioned in the article mainly refers to the process of college students returning to their hometown from the city where the university is located during the winter vacation in 2019. These students live in cities, towns or rural areas. At the same time, in the article, we selected two important variables: the student's household registration (rural, township, city) and residence during the epidemic (Wuhan, Other cities and counties in Hubei Province, Other provinces), and compared these two variables separately differences in infected student. It can be seen from the regression analysis model 1 in Table 5:
(1) Page 9 Line 485-487: There were significant urban-rural differences in the infection rate, namely, the infection rate for students with an urban registration (city) was higher than that of students with a rural registration by 181% (95% CI [51%, 421%], p<0.05).
(2) Page 11 Line 519-524: Model 1 showed that the risk of infection among students living in Wuhan during the pandemic was significantly higher than that of students living in the outside of Wuhan. The risk of infection among students living in other cities and counties in Hubei Province was only 0.25 times that of those living in Wuhan (95% CI [0.23, 0.50], P<0.001), and it was only 0.18 times in other provinces than those living in Wuhan (95% CI [0.1, 0.33], p<0.001).
- Please clarified also what are the symptoms that you use to classified mild, severe and critical cases.
Response: please refer to the revised contents of the first question of Reviewer 1.
- Discussion section is too long. The Authors some times repeat the same concept. Please reduce this section.
Response: Thank you very much for your suggestion, we have made the discussion part more concise. As stated on Page 11-13(Please refer to the content highlighted in yellow in the delete tag).

Reviewer 3 Report
This study uses survey data from the end of April 2020 to analyze the status of confirmed COVID-19 cases, the group differences, and influencing factors in college students in Wuhan. The sample size was made up 4355 participants, including 70 students with confirmed COVID-19 infection; 1.61 %, which is lower than previously reported in other countries (e.g.: https://www.tandfonline.com/doi/full/10.1080/10255842.2020.1869221). However, this study showed that the infection rate for college students was low, but it is necessary to prevent them from becoming asymptomatic carriers and thus spread the virus during traveling; furthermore, prevention and control measures for colleges should focus on maintaining social distancing, controlling population aggregation, and reducing the population density.
General comment
As expressed by authors, study has some important limitations. They are based on self-assessment and no clinical validation is performed. Students may hide their illness, authenticity/accuracy of data is not confirmed and the results for time of disease onset may be biased. Due to the above limitations, caution should be used when interpreting some results of this study (such as the infection rate).
Can the authors show the results in the seven-day per 100,000 incidence? It is a powerful metric to characterize the outbreak dynamics of COVID-19. It smoothes fluctuations in weekly reporting and scales case numbers to a fixed, easy-to-compare population.
Specific points
- Additional literature may be required in the introduction: Examining trends in COVID-19 transmission with case studies from Exeter and Loughborough universities, and research from other higher education institutions in England: https://www.ons.gov.uk/peoplepopulationandcommunity/educationandchildcare/articles/howhascoronaviruscovid19spreadamongstudentsinengland/2020-12-21 More studies: https://www.ncbi.nlm.nih.gov/pmc/articles/PMC7537562/ and https://www.ncbi.nlm.nih.gov/pmc/articles/PMC7537557/
- Source of literature is not adequate by using Sci-Hub. Better to search in Pubmed.
- In methods, time to respond the survey seems to be very low (only three days?).
- Correct table 1. COVID-19 infection is wrong. The 70 students should be referred as "yes" and the other way around.
- Not really clear in the results when authors claim that infection is higher in Ministry of Education-dependent Colleges. Can the authors show those infected people's hometown? Can you infer whether they got infected in the college or after leaving Wuhan? Not clear in the text.
- Not clear statistical differences between CPC members and non-CPC members in infection rate. Why is this variable relevant?
Author Response
Review report form from Reviewer3
This study uses survey data from the end of April 2020 to analyze the status of confirmed COVID-19 cases, the group differences, and influencing factors in college students in Wuhan. The total sample size was made up 4355 participants, including 70 students with confirmed COVID-19 infection; 1.61 %, which is lower than previously reported in other countries (e.g.: https://www.tandfonline.com/doi/full/10.1080/10255842.2020.1869221). However, this study showed that the infection rate for college students was low, but it is necessary to prevent them from becoming asymptomatic carriers and thus spread the virus during traveling; furthermore, prevention and control measures for colleges should focus on maintaining social distancing, controlling population aggregation, and reducing the population density.
General comment
- As expressed by authors, study has some important limitations. They are based on self-assessment and no clinical validation is performed. Students may hide their illness, authenticity/accuracy of data is not confirmed and the results for time of disease onset may be biased. Due to the above limitations, caution should be used when interpreting some results of this study (such as the infection rate).
- Can the authors show the results in the seven-day per 100,000 incidence? It is a powerful metric to characterize the outbreak dynamics of COVID-19. It smoothes fluctuations in weekly reporting and scales case numbers to a fixed, easy-to-compare population.
Response: We are very sorry, because this is a sample data that we cannot process it into an analysis result based on the population data.
Specific points
- Additional literature may be required in the introduction: Examining trends in COVID-19 transmission with case studies from Exeter and Loughborough universities, and research from other higher education institutions in England: https://www.ons.gov.uk/peoplepopulationandcommunity/educationandchildcare/articles/howhascoronaviruscovid19spreadamongstudentsinengland/2020-12-21 More studies: https://www.ncbi.nlm.nih.gov/pmc/articles/PMC7537562/ and https://www.ncbi.nlm.nih.gov/pmc/articles/PMC7537557/
- Source of literature is not adequate by using Sci-Hub. Better to search in Pubmed.
Response: Thank you for your reference materials. We have added relevant content in the introduction to the paper. Please refer to the revised contents of the third question of Reviewer 2.
- In methods, time to respond the survey seems to be very low (only three days?).
Response: We completed this survey within three days, because the survey is a task of the Ministry of Education, in order to assess the physical and mental conditions of teachers and students in Wuhan during the epidemic. First, it is not a monitoring investigation. The COVID-19 epidemic in Wuhan and China was basically under control in April 2020. Wuhan was unblocked on April 8. We conducted this retrospective investigation at the end of April. Second, our investigation received support and cooperation from the Hubei Provincial Department of Education and major universities in Wuhan, so the investigation was completed within a short period of time. However, we have maintained preparations for more than ten days, including formulating sampling frames, designing survey manuals, and guiding the selected college student management departments on how to conduct sampling.
- Correct table 1. COVID-19 infection is wrong. The 70 students should be referred as "yes" and the other way around.
Response: Thank you very much for your reminder, and we have modified the typos as NO/YES in Table 1(Page 4, Line 334).
- Not really clear in the results when authors claim that infection is higher in Ministry of Education-dependent Colleges. Can the authors show those infected people's hometown? Can you infer whether they got infected in the college or after leaving Wuhan? Not clear in the text.
Response: There are several reasons to explain that the infection is higher in Ministry of Education-dependent Colleges. On the one hand, Colleges which are directly affiliated with the Ministry of Education generally have late holidays, mostly in mid-to-late January2020. On the other hand, it can be seen from Table 4 that the higher proportion of infections are Master and PhD students (41%). These students are mainly from colleges and universities directly under the Ministry of Education. They will leave school and go home later because of experiments or research work. According to Figure 1, the later the date of departure from Wuhan, the significantly higher the infection rate was.
Regarding the location of infection, according to the date of infection in Figure 1, we speculate that most of the students who were diagnosed should be infected after they left the campus during holidays, including on the way of home, especially on train stations and trains. Because after the outbreak in January 2020, Wuhan university students are unlikely to be infected in their hometowns, but they are more likely to become the spreaders of the outbreak in their hometowns. The Chinese government launched a nationwide epidemic control on January 23, 2020--all the population from Wuhan, including university students in Wuhan, were quarantined.
- Not clear statistical differences between CPC members and non-CPC members in infection rate. Why is this variable relevant?
Response: We believe that party membership is an important characteristic of Chinese college students, and we incorporate it into the model with sociodemographic characteristics such as gender and age. In table 5(Page 9, Line 333-335), the infection rate for CPC members was 0.37times higher than that for non-CPC members (95% CI [0.19, 0.71], p<0.05). Why the infection rate of CPC members is lower than that of non-CPC members? In Chinese universities, students who are usually excellent in character and academics can become CPC members. We believe that we can analyze from the behavioral habits and believe that CPC member students are more cautious in behavior and their scope of activities outside campus is smaller. This may be consistent with the results of gender. In the regression model, the infection rate of girls is also significantly lower than that of boys.

Round 2
Reviewer 2 Report
Dear Authors,
Thank you very much for your job.